# Neutrophil Elastase Targets Select Proteins on Human Blood-Monocyte-Derived Macrophage Cell Surfaces

**DOI:** 10.3390/ijms252313038

**Published:** 2024-12-04

**Authors:** Nadia Tasnim Ahmed, Apparao B. Kummarapurugu, Shuo Zheng, Gamze Bulut, Le Kang, Aashish Batheja, Adam Hawkridge, Judith A. Voynow

**Affiliations:** 1School of Pharmacy, Virginia Commonwealth University (VCU), Richmond, VA 23298, USA; ahmedn12@vcu.edu; 2Children’s Hospital of Richmond at VCU, Richmond, VA 23298, USA; apparao.kummarapurugu@vcuhealth.org (A.B.K.); shuo.zheng@vcuhealth.org (S.Z.); gamze.bulut2013@gmail.com (G.B.); 3Department of Biostatistics, Virginia Commonwealth University (VCU), Richmond, VA 23298, USA; lkang@apps.vcu.edu; 4School of Medicine, Virginia Commonwealth University (VCU), Richmond, VA 23298, USA; bathejaa@vcu.edu

**Keywords:** neutrophil elastase, ELANE, cystic fibrosis, COPD, macrophages, polarization, inflammation, proteoglycans

## Abstract

Neutrophil elastase (NE) has been reported to be a pro-inflammatory stimulus for macrophages. The aim of the present study was to determine the impact of NE exposure on the human macrophage proteome and evaluate its impact on pro-inflammatory signals. Human blood monocytes from healthy volunteers were differentiated to macrophages and then exposed to either 500 nM of NE or control vehicle for 2 h in triplicate. Label-free quantitative proteomics analysis identified 41 differentially expressed proteins in the NE versus control vehicle datasets. A total of 26 proteins were downregulated and of those, 21 were cell surface proteins. Importantly, four of the cell surface proteins were proteoglycans: neuropilin 1 (*NRP1*), syndecan 2 (*SDC2*), glypican 4 (*GPC4*), and CD99 antigen-like protein 2 (*CD99L2*) along with neuropilin 2 (*NRP2*), CD99 antigen (*CD99*), and endoglin (*ENG*) which are known interactors. Additional NE-targeted proteins related to macrophage function were also measured including *CD40*, *CD48*, *SPINT1*, *ST14*, and *MSR1*. Collectively, this study provides a comprehensive unbiased view of selective NE-targeted cell surface proteins in chronically inflamed lungs.

## 1. Introduction

Alveolar macrophages play an important role in the immune response of the lung [1]. A normal acute-phase response involves the recognition of a pathogen or allergen and subsequent release of pro-inflammatory cytokines that summon neutrophils and polarize macrophages to the M1 phenotype. Once the pathogen is neutralized, macrophages are polarized to the anti-inflammatory M2 phenotype to fully resolve the inflammation. However, macrophage function is severely dysregulated in the chronically inflamed lungs of patients with cystic fibrosis (CF), chronic obstructive pulmonary disease (COPD), and bronchiectasis [2]. The dominant underlying pathology in these conditions is chronic infection and inflammation where neutrophils and macrophages remain in a vicious pro-inflammatory cycle that does not resolve, thereby leading to increased lung dysfunction and tissue damage. A key element of the pro-inflammatory phenotype is the sustained exocytosis of high-concentration proteolytic enzymes (e.g., Proteinase3, cathepsin G, and neutrophil elastase) during the degranulation of neutrophils [3,4]. Unraveling the complex roles that these enzymes play in perturbing innate immune cell function remains a major challenge.

Neutrophil elastase (NE or *ELANE*) is among the most abundant proteinases in neutrophil granules with an approximate cellular concentration of 5 mM [3,5]. NE is released into the airway milieu of CF patients at concentrations that can exceed 500 nM [6]. Once released into the airway milieu at sustained high concentrations, NE cleaves cellular and extracellular proteins impacting innate immunity [5,7], upregulating airway mucin *MUC5AC* expression, stimulating goblet cell metaplasia, inhibiting ciliary motility, degrading cystic fibrosis transmembrane conductance regulator (*CFTR*), and activating epithelial sodium channel (*ENaC*) [5]. The concerted effect of these activities increases airway mucus dehydration and obstruction and promotes infections. Excessive airway NE also impairs innate immunity by degrading innate immune proteins such as transferrin and lactoferrin, cleaving opsonins and opsonin receptors, cleaving the phosphatidyl serine receptor resulting in efferocytosis failure, blocking dendritic cell maturation, and generating fibrin degradation products that are chemotactic for neutrophils to perpetuate inflammation [5]. NE has pro-inflammatory effects in airway epithelial cells including activation of Toll-like receptor 4 (*TLR4*), high-mobility group box 1, the Interleukin 1 (*IL-1*) family of pro-inflammatory cytokines, and pre-proteases to further enhance protease load in the lung. In addition to promoting release of Interleukin 8 (*IL-8*) and Tumor Necrosis Factor α (*TNFα*), NE increases oxidative stress through the release of heme-free iron for cellular uptake and degradation of important anti-proteases including Tissue Inhibitor of Metalloproteinases 1 (*TIMP1*) and Secretory Leukocyte Protease Inhibitor (*SLPI*) (reviewed in reference [5]).

We have reported that extracellular NE has a significant impact on the sentinel immune cell in the lung, the macrophage. NE treatment causes macrophage phagocytic failure [7,8]. NE instillation into the airways of Sprague Dawley rats increases iron uptake into the macrophages in the bronchoalveolar lavage (BAL) and in the lung [9]. Human blood-monocyte-derived macrophages (BMDMs) take up exogenous NE intracellularly into the cytoplasm and nucleus where NE retains proteolytic activity [10]. NE activates the release of macrophage extracellular traps (METs) from healthy BMDMs, CF BMDMs [10], and COPD BMDMs [11]. Furthermore, NE proteolytic activity targets several intracellular proteins including histone H3 that serves as a precursor to chromatin decondensation and MET release [10], Histone Deacetylases (HDACs), and Sirtuin 1 [12]. Importantly, NE does not affect histone acetyltransferase activity, resulting in unopposed lysine acetyltransferase activity in macrophages post NE treatment [12]. We sought to use unbiased quantitative liquid chromatography–tandem mass spectrometry (LC-MS/MS) proteomics to further evaluate the global impact of NE on the BMDM proteome.

## 2. Results

### 2.1. Proteomics Analysis of NE-Treated vs. Control BMDMs

Label-free quantitative (LFQ) LC-MS/MS proteomics was used to study the effects of neutrophil elastase (NE) on the BMDM proteomes to identify proteolytic targets and early-stage cell signaling events (Figure 1). For this study, we focused on three healthy unidentified individuals (B23, B25, and B26) who donated blood at the American Red Cross. We isolated blood monocytes from the donor buffy coats and aliquoted equivalent BMDM populations into six-well plates for culture and macrophage differentiation. The culture methods required Granulocyte-Macrophage Colony-Stimulating Factor (GM-CSF) for 8–10 days to permit differentiation confirmed by CD11b+, by Wright Giemsa stain for morphology, and by response to stimulation with lipopolysaccharide (LPS) or Interleukin-4 (IL-4)/Interleukin-13 (IL-13), which confirmed the potential to differentiate to M1 or M2 phenotypes [10]. Following differentiation, the BMDMs were treated in triplicate with either control vehicle or 500 nM NE (+NE) for 2 h. Following a 2 h incubation, the primary cells were harvested and individually analyzed by LC-MS/MS.

A total of 5262 unique proteins were identified across all three donor samples of which we quantified 4560, 4813, and 4435 proteins from donors B23, B25, and B26, respectively (Figure 2A; Appendix A). Importantly, the replicate lysates from each donor were searched independently such that data between samples did not contribute to identifications, quantification, or statistical analysis. A summary of shared quantified proteins is provided in a Venn diagram (Figure 2B) wherein 3928 proteins were shared between the three donors. Statistical analysis of the relative quantification data was performed within each donor sample set (*t*-test, adjusted *p*-value) which resulted in the identification of 248, 163, and 164 statistically significant proteins (adj *p*-value < 0.05) from the B23, B25, and B26 donors, respectively (Figure 2C). Closer analysis of the statistically significant dataset showed that a total of 41 proteins were detected in all three donors and statistically significant in at least two out of three donors. The expression levels for these 41 proteins is reported in Figure 2D with insignificant relative protein levels indicated with an asterisk.

### 2.2. NE Targets Selected Cell Surface Proteins

The 41 proteins reported in Figure 2D were grouped into three categories: (1) upregulated, (2) downregulated, and (3) inconclusive. There were 8 proteins (~20%) that were upregulated including *ELANE* which was added in the treatment group, 26 proteins (~63%) that were downregulated, and the remaining 7 (~17%) were statistically significant yet showed up- and downregulation in different donor samples. We performed gene enrichment analysis of the 8 upregulated and 26 downregulated proteins in g:Profiler against a background of all quantified proteins from the study (N = 5262). No enriched gene ontology (GO) pathways were identified for the upregulated proteins which included *CBLB*, *CRLF3*, *CTPS1*, *ENOSF1*, *HSP90AA4P*, *IMP3*, and *PPP3CB*. Among the 26 downregulated proteins, *ENG* (*CD105*), *MMP14*, *NRP1*, *NRP2*, *SPINT1*, and *ST14* were enriched (adj *p*-value < 0.05) in GO biological processes involving morphogenesis of branching epithelium (GO:0001763), branching structure (GO:0061138), and outflow tract septum morphogenesis (GO:0003148). Furthermore, 20 out of the 26 downregulated proteins (~77%) were enriched (adj *p*-value < 0.05) in the following GO cell components as summarized in Figure 3A: cell surface (GO:0009986), cell periphery (GO:0071944), side of membrane (GO:00098552), and receptor complex (GO:0043235).

Approximately 25% of the total number of quantified proteins in this study are annotated as being located on the cell surface/plasma membrane, yet only a small fraction (<2%) of those proteins were targeted by NE and proteolytically degraded. To further support the evidence that NE is targeting specific proteins at the extracellular cell surface, we plotted the 26 downregulated proteins by LFQ abundance and rank in Figure 3B. The LFQ dataset for the 3928 shared proteins (Figure 2B) covers approximately 6 orders of magnitude and the 26 downregulated proteins are spread over the middle ~3.5 orders of magnitude of LFQ abundance, suggesting no bias towards more abundant proteins. Two labeled proteins of note are *SDC2* and *GPC4*, the latter plotted in red. *SDC2* is technically not annotated as a membrane protein, yet it is a well-established transmembrane protein. *GPC4* was significantly downregulated in donors B25 and B26 but not detected in donor B23 and therefore not included in Figure 2A or the gene enrichment analysis. However, its relevance will become apparent in subsequent discussions (vide infra).

### 2.3. NE Targets Proteoglycans

Proteoglycans are a unique class of proteins that host glycosaminoglycans (GAGs), a heterogeneous linear polysaccharide post-translational modification (PTM). There are fewer than ~100 proteins that are known to contain GAGs, yet we identified four in our proteomics analysis. These include *NRP1*, *SDC2*, *GPC4*, and *CD99L2* (recently identified as a proteoglycan) [13]. *NRP1*, *SDC2*, *GPC4*, and *CD99L2* are all cell surface membrane proteins and all, with the exception of *GPC4* in one donor sample, were significantly downregulated in the NE-treated BMDMs. Also of note was the presence of down-regulated proteins *CD99*, *ENG*, and *NRP2* which are co-receptors or ligands of the proteoglycans putting them in close proximity to enriched NE.

To confirm NE targets proteoglycans, we performed Western blots against *NRP1*, *SDC2*, and *GPC4* levels as a function of NE concentration in three human donor BMDM samples (Figure 4). BMDMs were treated without NE (Ctrl), 200 nM NE, and 500 nM NE for 1 and 2 h. The Western blot data for *NRP1* (Figure 4A) and *SDC2* (Figure 4C) showed clipping and degradation while the Western blot for *GPC4* (Figure 4E) showed overall decreased expression. These Westerns were performed from the same three donors and β actin Western confirms the integrity and equal loading of the protein samples for each Western. These data confirm the LFQ proteomics dataset and establish cell surface proteoglycans as a class of proteins degraded by NE. However, it is possible that the decrease in protein in cell lysates that was detected by Western and proteomics analyses may be due to release of the clipped or cleaved proteins into the conditioned media. That possibility remains to be tested.

## 3. Discussion

Neutrophil elastase has proteolytic specificity preferentially targeted to the C-terminal side of valine and alanine. NE has been shown to proteolytically target immune cell surface proteins in CF and COPD patients including *CD40*, *CD80*, and *CD86* on dendritic cells [14], *CD2*, *CD4*, and *CD8* on T cells [15], and *CD14*, *CD206*, and *CD44* on human BMDMs [16,17]. Herein we report the use of LC-MS/MS proteomics to measure in an unbiased manner the NE-mediated effects on BMDMs. A total 5262 proteins were identified including approximately 1300 proteins that were annotated as membrane-associated proteins providing an unprecedented view of cell surface protein dynamics in the presence of high NE levels. Interestingly, only 41 proteins were significantly dysregulated after 2 h of 500 nM NE treatment in at least two out of the three donor samples. When we considered the 26 downregulated proteins, we found that 21 were cell surface proteins which represents <2% of the ~1300 cell surface proteins quantified in this study. Out of the 21 cell surface proteins in our study, the 3 that agreed with previous studies were *CD40* [14], *CR1* [18], and *MMP-14* [16]. Additional cell surface proteins that have been previously reported to be targeted by NE such as *CD14* [16,17,19] *CD2*, *CD8*, and *CD4* [15] were either not detected in our study or detected but not found to be significant.

A significant finding in this study was that NE selectively targets membrane bound proteoglycans *NRP1*, *SDC2*, *CD99L2*, and *GPC4*. Proteoglycans are a unique class of glycoproteins that contain glycosaminoglycans (GAGs). Unlike N- and O-linked glycans that have branched structures, GAGs are linear polysaccharides with large polydispersity (10–100 kDa) and chemical heterogeneity. GAGs are comprised of four main types: (1) heparin/heparan sulfate, (2) chondroitin sulfate, (3) keratan sulfate, and (4) hyaluronan [20]. These physiochemically heterogeneous polysaccharides with a high density of anionic sulfate charges bind with high affinity to NE [21,22]. Proteoglycans regulate monocyte, dendritic cell, and macrophage function by regulating leukocyte extravasation, and by serving as binding factors for cytokines, chemokines and growth factors [20]. Previous work by Campbell et al. [21] showed that heparan sulfate and chondroitin sulfate bound extracellular NE on the surface of neutrophils. A key conclusion was that GAG-bound NE remained proteolytically active but was shielded from proteolytic inhibitors such as *SLP1* and α-1-antitrypsin. Consistent with this observation, human leukocyte elastase (NE) also binds to soluble high-Mr heparin sulfate proteoglycans (including syndecans-1 and -4) shed into bronchial secretions from patients with bronchiectasis [23] and NE bound to these syndecans retains its catalytic activity against extracellular matrix substrates and is protected from α-1-antitrypsin [23]. In contrast, sputum NE bound to soluble heparin sulfate groups or DNA in CF sputum results in partial inhibition of soluble HLE activity in vitro [22,24] due to binding of negatively charged molecules to the allosteric activation site of HLE, causing protease inhibition [22]. In summary, NE shedding of glycosaminoglycans from BMDM cell surfaces may impact elastase activity and susceptibility to antiproteases.

*NRP1* and *NRP2* are transmembrane proteins that can form homo and heterodimers [25]. Both proteins play crucial roles in axonal development, angiogenesis, cancer progression, and immune cell development, migration, recruitment, and immune response [26,27]. In the macrophage, *NRP1* is localized to the cell membrane and regulates M2 polarization and tumor-associated macrophage function [27]. *NRP1* and *NRP2* serve as co-receptors for diverse growth factors including *SEMA3* [28,29] and *VEGFA_165_* [30,31]. *NRP1* is post-translationally modified with both chondroitin and heparan sulfate in a tissue- and cellular-specific manner [30,32], whereas *NRP2* is not known to contain the GAG modifications of *NRP1*. Loss of *NRP1* enhances cigarette-smoke-induced emphysema [33], facilitates NE uptake in breast cancer cells and resulting cytotoxicity [34,35], and serves as a co-receptor for *ACE-2* during COVID-19 infection [36]. Importantly, consistent with our proteomic results, a prior study evaluating the impact of NE on shed and secreted proteins from cultured RAW264.7 murine macrophage cells revealed that both *NRP1* and *NRP2* are present in the conditioned media following NE treatment but not after control treatment conditions [8].

*SDC2* is a versatile heparan sulfate proteoglycan involved in various cellular functions and implicated in cancer biology and inflammation [37,38]. It features a complex structure with a protein core and glycosaminoglycan (GAG) chains [37]. *SDC2* expression is modulated in various health conditions. *SDC2* is required for bacterial clearance [38]. *SDC2* is upregulated in the lungs of smokers [39]. It interacts with growth factors like *TGFβ*, influencing signaling pathways and cellular adhesion [40]. Previous studies have shown that *SDC1* interacts with NE in airway inflammation [24].

*CD99L2* is one of the newer proteoglycans, having recently been discovered to contain a chondroitin sulfate modification on Ser_178_ in the extracellular domain [13]. In addition to *CD99L2*, we also measured downregulation of *CD99* in all three donor samples in the NE treatment sets. *CD99* antigen [41,42] and *CD99* antigen-like protein 2 (*CD99L2*) are related transmembrane proteins that form heterodimers [43] and play important roles in leukocyte extravasation and cancer. Interestingly, *CD99* ligation with a monoclonal antibody induces reprogramming of M0 and M2 macrophages to the M1 pro-inflammatory phenotype [44]. Rutledge et al. [45] showed that *CD99L2* regulates leukocyte migration and *CD99L2*-deficient mice have defective inflammatory responses. Their study revealed that *CD99L2* regulates transendothelial migration of leukocytes by operating sequentially with platelet endothelial cell adhesion molecule (PECAM, *CD31*) and *CD99*. Specifically, *CD99L2* recruits the lateral border recycling compartment downstream of PECAM, highlighting its unique role in inflammation [45].

Glypican-4 (*GPC-4*) is a heparan sulfate glycoprotein anchored to the cell membrane by a glycosylphosphatidylinositol (GPI) anchor. It regulates cell growth, migration, and differentiation [46]. Loss of *GPC*-*4* is associated with breast cancer metastases [46]. *GPC-4* regulates *Wnt3a* and *Wnt5a* signaling and activation of *β-catenin* [47]. *GPC-4* is also associated with inflammation; it is upregulated on the surface of macrophages in rheumatoid arthritis [48].

Several cell surface proteins which were downregulated are associated with M1 macrophage polarization and pro-inflammatory function. *CD40* is a monocyte/macrophage cell surface protein which plays a role in macrophage immunosurveillance. When activated by the ligand *CD154* on activated T cells, *CD40* recruits TNFR Activating Factor to activate pro-inflammatory responses to infections [49]. Another pro-inflammatory receptor, macrophage scavenger receptor 1 (*MSR1*), is associated with increased susceptibility to COPD [50]. One *MSR1* SNP, P275A, is associated with macrophage cell survival, adhesion, and receptor expression, which in turn influences COPD-related lung damage and inflammation [50]. *MSR1* expression is significantly increased in peripheral blood mononuclear cells (PBMCs) from patients with asthma and COPD [50]. *SPINT1* is a hepatocyte growth factor activator inhibitor-1 (HAI-1) that promotes M1 macrophages and inhibits M2 macrophages, resulting in improved outcomes for patients with non-small cell lung carcinoma [51].

In contrast, several cell surface proteins downregulated by NE promote M2 polarization or tumor-associated macrophage function and cancer progression. *ENG* is a single-pass membrane protein that is part of the TGF-beta receptor complex [52]. *ENG* has been shown to play an important role in macrophage differentiation and polarization [53]. Specifically, *ENG* affected macrophage senescence and transfected isoforms of *ENG* (short *ENG*) in U937 cells showed increased levels of M2-like genes. Bone marrow stromal cell antigen 2 (*BST2*) is a type II transmembrane protein expressed on both tumors and leukocytes including macrophages. The presence of *BST2* on macrophages is associated with increased colorectal carcinoma progression and tumor-associated macrophage (TAM) phenotype (M2 phenotype) infiltration. Silencing *BST2* results in restrained CRC progression and change in macrophage polarization to M1 [54]. In pancreatic ductal adenocarcinoma (PDAC), *BST2*+ macrophages induce CD8+ T cell exhaustion via activation by ERK/*CXCL7*, resulting in PDAC tumor growth [55]. Plexin Domain Containing 2 (*PLXDC2*) deficiency in bone-marrow-derived macrophages (BMDMs) is associated with increased inflammation, as demonstrated by its role in modulating the host immune response during *Helicobacter pylori* infection and in chemically-induced colitis models [56]. Conversely, overexpression of *PLXDC2* in stromal-associated M2 macrophages is linked to epithelial-to-mesenchymal transition (EMT) and gastric cancer progression. This overexpression correlates with poor survival, advanced tumor stage, and heightened M2 macrophage presence, further contributing to inflammatory signaling pathways and the EMT process [57].

Neutrophil elastase treatment resulted in the upregulation of seven proteins, three of which regulate macrophage inflammatory signaling or inhibit tumor cell cycle progression. One protein, Casitas-B-lineage lymphoma protein-b (*CBL-b*), a CBL-family E3 ubiquitin ligase, is expressed on macrophages and inhibits Th2/Th9 and allergic airway disease by targeting STAT6 for ubiquitination. Importantly, *CBL-b* also restrains Th17 expression and activity on CD4+ T cells by suppressing IL-6 expression. Therefore, *CBL-b* suppresses both Th17 and Th2 inflammation [58]. Two other proteins (*CTSP1* and *PPP3CB*) are protein phosphatases which suppress activation of cancer cells. Complete S Transactivated protein 1 (*CTSP1*) is a protein phosphatase that dephosphorylates *Akt* at Ser473 which blocks bladder cancer cell cycle progression and promotes cell apoptosis. *CTSP1* suppresses bladder cancer growth [59]. Protein phosphatase 3 catalytic subunit beta (*PPP3CB*) is a Ca/Calmodulin-dependent serine/threonine protein phosphatase expressed in the brain. A malignant brain tumor, glioblastoma, is associated with low expression levels of *PPP3CB*. *PPP3CB* expression is associated with activation of immune cells which are present in the tumor and expression of immune checkpoint genes to suppress uncontrolled tumor cell growth [60]. In contrast to these studies of NE-regulated suppression of cancer, ELANE is expressed in promyelocytic leukemia (PML) and sustains PML by inhibiting apoptosis via degradation of BAX a pro-apoptotic protein and upregulation of Bcl-2, an apoptosis inhibitor [61].

This study focused on NE-induced changes in the proteome and not the transcriptional or signaling changes activated by NE. Importantly, NE transcriptional activity is suppressed in macrophages likely due in part to Histone H3 clipping that removes the H3 N-terminus required for post-translational modifications for transcriptional activity [62]. Furthermore, macrophage treatment with NE alone does not upregulate the cytokine TNFα; macrophage upregulation of TNFα requires only LPS. However, the combination of macrophage treatment with LPS plus NE significantly increases both mRNA and protein expression of TNFα greater than LPS alone [16]. Therefore, NE regulation of the macrophage transcriptome is complicated and usually reflects co-exposures with other stimuli.

While this study provides insight into NE’s role in modulating macrophage surface proteins, it has certain limitations. We did not use an inactivated NE (PMSF-treated) or a catalytically inactive NE control, which would help confirm that the observed effects are due to NE’s proteolytic activity rather than other potential interactions. Future studies should include these controls to delineate NE-specific degradation mechanisms. Furthermore, we did not examine NE-induced signaling pathways or gene transcription changes that could provide additional insight into NE’s impact on macrophage responses. Investigating these pathways could offer a deeper understanding of NE’s downstream effects on macrophage polarization and functional responses. We examined the total cell lysate proteome but we did not analyze the conditioned media proteome. As we discussed above, other studies have confirmed some of our results in conditioned media from NE-treated BMDM cells or macrophage cell lines [8]. Our study was also performed in BMDM from healthy individuals. In future studies, we will evaluate the effects of NE on healthy and diseased BMDMs as well as investigate the effect of NE treatment time to allow cells time to adapt to chronic NE exposure. Lastly, while we observed significant proteolytic effects on cell surface proteins, we did not assess macrophage functional responses including cytokine production, phagocytic activity or chemotaxis post NE-treatment. Future research should address these functional outcomes to better elucidate the implications of NE treatment on macrophage behavior in an inflammatory environment. By integrating these additional analyses, future work can expand on the current findings to clarify NE’s multifaceted role in immune modulation and its potential therapeutic implications in inflammatory diseases.

## 4. Materials and Methods

### 4.1. BMDM Culture and Treatment

Buffy coats from deidentified healthy donors (American Red Cross) were processed by a modified two gradient method using Lymphoprep and Percoll gradients [63]. Isolated blood monocytes were counted and adhered to 6-well plates and differentiated in RPMI growth media with GM-CSF (20 ng/mL, BioLegend, #572904, San Diego, CA, USA) over 8–10 days. On the day of treatment, BMDM were washed twice with PBS and incubated with control vehicle or 500 nM NE (Elastin Products, SE563, Owensville, MO, USA) in serum-free RPMI treatment media for 2 h. At the end of treatment, the cells were washed 3× with 1 mL PBS pH 7.4, harvested by scraping in 1 mL of PBS (pH 7.4) with HALT protease inhibitor (ThermoFisher, 78438, Waltham MA, USA) pelleted in 1.5 mL centrifuge tubes, and then snap frozen and stored at −80 °C.

### 4.2. Proteomics Sample Preparation

The samples were removed from the −80 °C storage and processed using the iST 8x sample kit (PreOmics, Seattle, WA, USA) according to the manufacturer’s instructions. Briefly, samples were thawed for 5 min at room temperature followed by the addition of 50 μL of lyse buffer from the iST kit. Each pellet was lysed using an ultrasonic probe (10 cycles @30 s each) on ice followed by heating at 95 °C for 5 min on an orbital shaker. A BCA assay (ThermoFisher, 23225) was used to measure the total protein for all samples. A total of 100 µg of total protein from each sample was aliquoted into separate iST cartridges and digested (LysC/trypsin) at 37 °C on a shaker for 2 h. After incubation, 100 μL of Stop solution was added, the samples were centrifuged at 3800 rcf for 3 min, and they were then subjected to subsequent centrifugation wash steps involving 100 μL of Wash 1, 100 μL of Wash 2, and 100 μL of Elute solution. was added to each cartridge and centrifuged at 3800 rcf for 3 min and then repeated eluted peptide samples were dried in a speedvac at 45 °C for 1.5 h and stored at −20 °C. Prior to LC-MS/MS analysis, the dried samples were reconstituted in 50 μL of LC-Load solution and a peptide quantification assay (ThermoFisher, 23290) was used to measure the total peptide concentration for all samples. The final volume adjusted to provide a total peptide concentration of 500 ng/μL.

### 4.3. LC-MS/MS

Proteomics measurements were made using a Thermo Fisher Q-Exactive HF-X tandem mass spectrometer coupled with an nLC 1200 UHPLC system equipped with an Easyspray ion source. The nLC set-up consisted of an Acclaim PepMap 75 μm × 2 cm nanoviper C18 3 μm × 100 Å pre-column in series with an Acclaim PepMap RSLC 75 μm × 15cm 2 μm C18 analytical column and the ESI source set to 1.5 kV. A total of 500 ng (1 µL) of peptide digest was injected onto the column per sample and eluted at a flow rate of 300 nll/min for 131 min with a linear gradient consisting of mobile phases A (100% H_2_O, 0.1% formic acid) and B (80% ACN, 0.1% formic acid). The gradient used was as follows: 0–5 min (2% B), 5–105 min (30% B), 105–125 min (50% B), 125–126 (100% B), and 126–131 (100% B). The LC-MS/MS system was operated in a data-dependent acquisition mode with a full scan mass spectrum (350–1575 *m*/*z*; 120,000 RP, AGC = 3 × 10^6^, and IT_max_ = 100 ms) followed by 15 MS/MS spectra (30,000 RP, AGC = 1 × 10^5^, IT_max_ = 50 ms, and NCE = 28).

### 4.4. Western Blot Analyses for Validation of LC-MS/MS Identified Protein Targets

Human BMDMs obtained from healthy donors were seeded onto 6-well plates and treated with NE (200 and 500 nM) or control vehicle for 1 and 2 h at 37 °C. Following NE exposure, equal amounts of total cell lysates (40 µg) were separated by SDS-PAGE (4–12%). After blocking with 5% milk in PBS-T (0.1% Tween 20 in PBS) for 2 h at room temperature, membranes were incubated with rabbit anti-neuropilin 1 (1:1000, #54568; CST), rabbit anti-syndecan 2 (1 µg/mL, #36-6200; Invitrogen, Waltham, MA, USA), and rabbit anti-glypican 4 (1:2000, PA5-88766; Invitrogen) antibodies. This was followed by incubation with goat anti-rabbit-HRP conjugated antibody (1:5000; #7074, Cell Signaling Technology, Danvers, MA, USA). Immunoreactive protein complexes were detected with SuperSignal West Pico Chemiluminescent Substrate (Cat #34580, Thermo Fisher Scientific). To confirm equal loading, blots were re-hybridized with mouse monoclonal antibody specific to β-actin (1:6000, A5441, Sigma-Aldrich, St. Louis, MO, USA) and secondary HRP-conjugated goat anti-mouse IgG. Protein bands for full-length NRP1, SDC2, GPC4, and β-actin were quantified to estimate intensity changes using ImageJ software (https://imagej.net/). Relative changes in each protein target band density were first normalized to β-actin and then presented relative to their corresponding control-vehicle-treated sample.

### 4.5. Statistical Analysis

Statistical significance for differential expression was determined through *t*-tests with false discovery rates calculated for each patient dataset using the Benjamini–Hochberg method. Proteins with adjusted *p*-values < 0.05 were considered significant. For Western blot densitometric measurements of NRP1, SDC2, and GPC4, data were analyzed using a one-way nonparametric ANOVA (Kruskal–Wallis test) with Tukey’s multiple comparisons test, performed in Prism software (https://www.graphpad.com/) (GraphPad, La Jolla, CA, USA).

### 4.6. Data Analysis

LC-MS/MS raw files were grouped by donor and searched in Proteome Discoverer 3.0 using CHIMERYS. Proteins with abundances in <5 of the LC-MS/MS runs were removed and not used for further LFQ analysis. The ratios for LFQ control/NE treatment (Ctr/NE) replicates were calculated for proteins in each patient, the median Ctr/NE ratios in each patient were log_2_ transformed, and then the median log_2_ (Ctr/NE) values were *t*-tested against the entire median log_2_ (Ctrl/NE) values for each patient. False discovery rates were calculated for each patient dataset (Benjamini–Hochberg) and adjusted *p*-values < 0.05 were considered significant. Proteins detected in all three donors and found with significantly altered LFQ levels in ≥two donors were considered for further bioinformatics analysis. Enrichment analysis was performed in g:Profiler [64] for significant proteins against a background of all quantified proteins in this study with the Organism: Homo sapiens, g:SCS threshold < 0.05, and using all available data sources but only highlighting the GO driver terms.

## 5. Conclusions

In summary, NE exposure decreased BMDM cell surface receptor expression which was the predominant class of proteins affected by NE exposure. Our results suggest that loss of these cell surface proteins generally appeared to suppress macrophage polarization and leukocyte activation. Interestingly, three of the seven increased proteins suppress inflammation or inhibit tumor cell proliferation and promote apoptosis. Furthermore, our results revealed the surprising finding that surface proteoglycans are a highly represented target for NE release/degradation from BMDM, suggesting that NE may be sequestered at these concentrated sites of negatively charged proteoglycans. The concerted loss of proteoglycans by NE exposure potentially degrades macrophage orchestration of inflammatory responses.

## Figures and Tables

**Figure 1 ijms-25-13038-f001:**
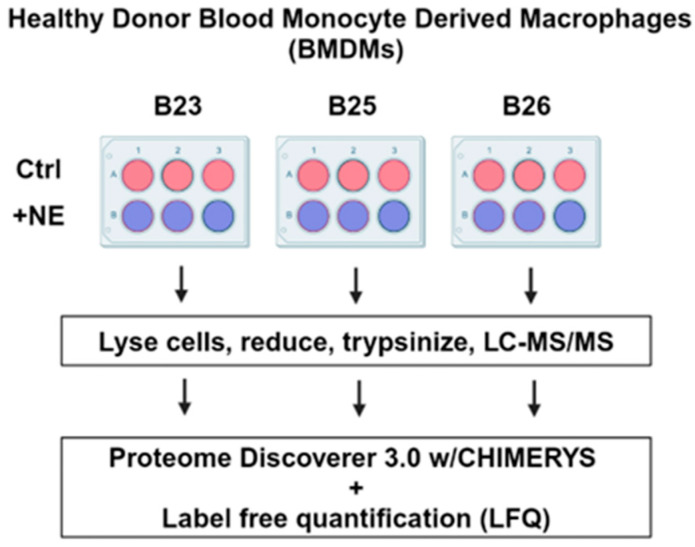
Experimental workflow for investigating the effects of NE treatment on the human blood-monocyte-derived macrophage (BMDM) proteome. BMDMs from healthy donors were split equivalently into 6-well plates and then incubated without (control (Ctl)) or with 500 nM of neutrophil elastase (+NE) for 2 h in triplicate.

**Figure 2 ijms-25-13038-f002:**
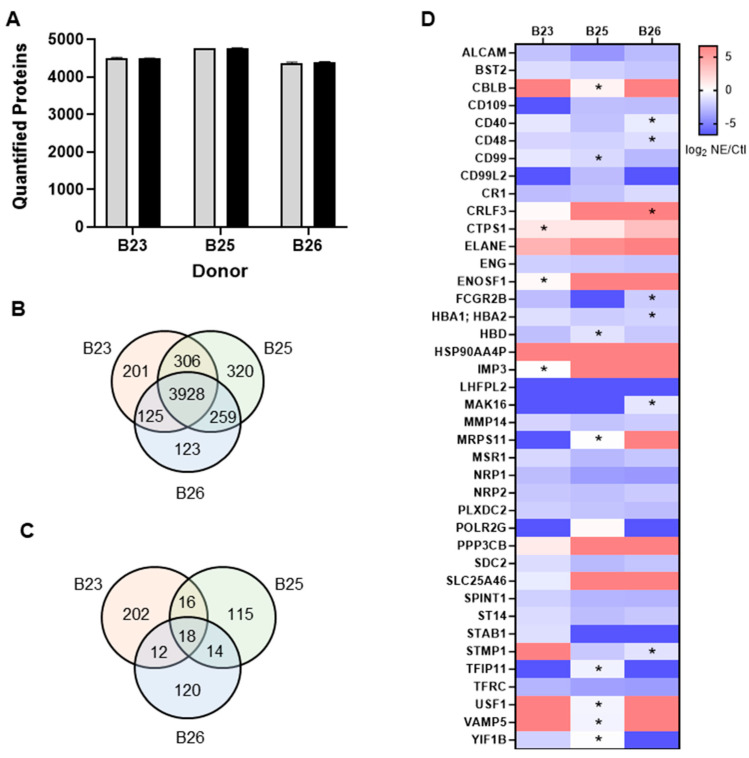
Summary of label-free quantification (LFQ) results. Results show the number of proteins quantified in each donor sample with NE treatment (black) and without NE treatment (gray) (**A**). Venn diagram shows the proteome overlap between donor samples (**B**). Venn diagram shows statistically significant (*t*-test; adjusted *p*-value < 0.05) differentially regulated proteins in NE vs. Ctl (**C**). Heatmap showing the log2 fold-change values of protein levels in NE vs. control that were detected in 3/3 donors and statistically significant in 3/3 or 2/3 donor (* indicates the expression level is not statistically significant) (**D**).

**Figure 3 ijms-25-13038-f003:**
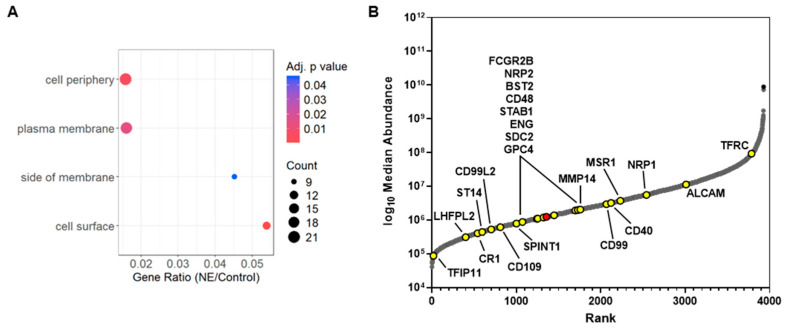
Down-regulated proteins. Enriched downregulated proteins from the GO cellular component annotation (**A**). Log10 of the median normalized LFQ abundances rank-plotted with 21 cell-membrane-annotated differentially downregulated proteins (yellow) labeled by gene. *GPC4* is labeled in red as it is downregulated in only 2/3 donor samples yet is a member of the proteoglycan family (**B**).

**Figure 4 ijms-25-13038-f004:**
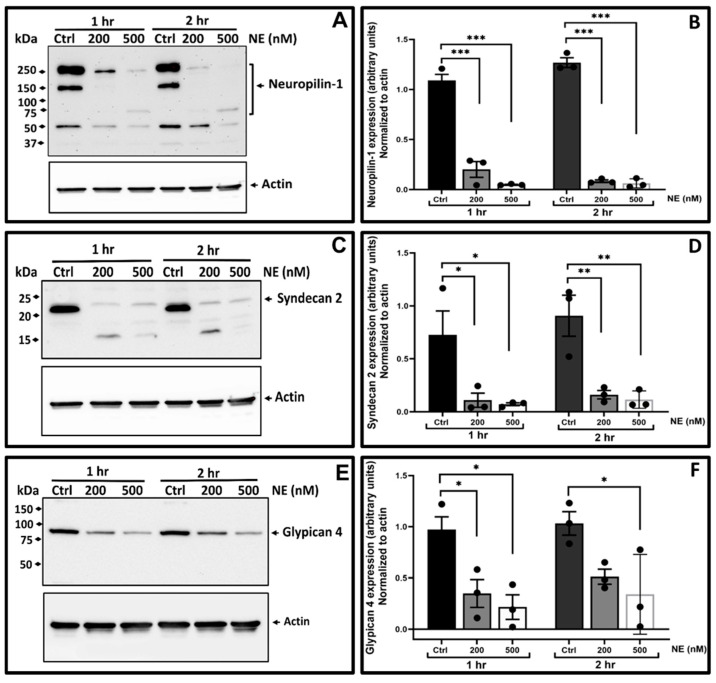
Western blot and densitometry results for *NRP1* (**A**,**B**), *SDC2* (**C**,**D**), and *GPC4* (**E**,**F**) from NE-targeted BMDM lysates. Three separate blood monocyte donations were obtained from healthy donors and cultured in RPMI with GM-CSF for 8–10 days to differentiate cells to blood monocyte derived macrophages. Cells from each donor were treated with control vehicle or NE (200 or 500 nM) for 1 or 2 h. Cell lysates were collected, protein quantified, and Western analyses were performed for *NRP1* (**A**), *SDC2* (**C**), and *GPC4* (**E**). Left panels are representative Western blots for protein targets and as a control, β-actin. After densitometry of bands using ImageJ, relative expression corrected for β-actin was compared to control treated samples and summary data of relative expression for each protein shown in the panel on the right, *NRP1* (**B**), *SDC2* (**D**), and *GPC4* (**F**). Significant differences in relative expression for N = 3 per protein was determined; *, *p* < 0.05; **, *p* < 0.01; ***, *p* < 0.001.

## Data Availability

The original contributions presented in the study are included in the article and Appendix A, further inquiries can be directed to the corresponding authors.

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
