# Peer review of "Neutrophil Elastase Targets Select Proteins on Human Blood-Monocyte-Derived Macrophage Cell Surfaces"

_ijms, 2024, doi:10.3390/ijms252313038_

Round 1

Reviewer 1 Report

Comments and Suggestions for Authors

Review of the manuscript entitled: Neutrophil elastase targets select proteins on human blood monocyte-derived macrophage cell surfaces.

The Authors' research concerns an important medical problem. The authors' research concerns an important medical problem. Research can contribute to, for example, understanding the mechanism of spreading leukemia cells in the body and the spread of lymphocytes during the development of inflammation. In my opinion manuscript is interesting but some corrections should be made before publication.

1.      The abstract and introduction are prepared mostly correctly. From technical notes, the aim of the manuscript should be clearly indicated in the abstract and at the end of the introduction e.g. "The aim of the present study was to ...". Moreover, in the introduction we do not describe the results. If the result of the work is described, why read further. Please correct the last paragraph of the introduction.

2.      All acronyms and abbreviations should be explained when they are first used e.g. line 61 “IL-8 and TNFα” or 62 “TIMP1 and SLPI” and similar, check the entire manuscript carefully.

3.      Generally results are described correctly. However, according to the requirements of current nomenclature gene names should be written in italics. This is crucial, correct throughout the manuscript also in figures.

4.      There is no need for additional subsections in the discussion. Please remove from the line 218, 242, 284, and 355.

5.      It is now known that elastin derived peptides (EDPs) affect COPD and macrophages. Moreover, neutrophil elastase contributes to the production of EDPs. Has this aspect been studied? maybe it would be worth discussing? The roles of EDPs and macrophages or leukemias and ELANE? Did the authors study proteins related to matrix production, e.g. elastin, collagen, etc.

6.      Lines 379-388 these are conclusions, they should be highlighted as a separate chapter. It is good practice to raise the limits of the work and possible perspectives of future research - if possible add.

7.      Did the research conducted have the consent of the bioethics committee? If so, provide the consent number.

8.      In materials and methods, a subsection on statistical analysis should be added.

Reviewer 2 Report

Comments and Suggestions for Authors

The authors confirmed and identified new protein targets of neutrophil elastase (HE) on blood monocyte-derived macrophages. Most protein targets were cell surface glycoproteins. It is a descriptive study. It provides no clues on how HE impacts macrophage functions, including macrophage reprogramming. The discussion is speculative and mainly a literature review on proteins contributing to macrophage polarization or other types of leukocytes (T lymphocytes). I suggest shortening the discussion section.

Other issues:

Page 5, lines 169-172: Authors discussed a few proteins (SRGN & AGRIN). These proteins are not in the figures. Although the authors refer to supplementary Table 1 and Supplemental Figure S1, the files were not annexed. The reader cannot verify the author's claims.

Page 5, lines 182-184. The authors conclude that macrophage treatment with HE induces a complete degradation of NRP1. I don't think they can say that. It simply means that proteolysis of NRP1 by HE removes the epitope recognized by the monoclonal antibody. The fragment recognized by the antibody is likely in the conditioned medium of HE-treated macrophages.

The authors have not assessed the functional responses of HE-treated macrophages. It is another major limitation.

HE is a protease. Why not use inactivated (PMSF treated) or a catalytically inactive HE as a control? What about HE-induced signaling and gene transcription known?

Round 2

Reviewer 2 Report

Comments and Suggestions for Authors

I have no additional comments.